# Basis of PD1/PD-L1 Therapies

**DOI:** 10.3390/jcm8122168

**Published:** 2019-12-08

**Authors:** Barbara Seliger

**Affiliations:** Institute for Medical Immunology, Martin Luther University Halle-Wittenberg, 06112 Halle (Saale), Germany; barbara.seliger@uk-halle.de; Tel.: +49-345-557-1357; Fax: +49-345-557-4055

**Keywords:** immune checkpoints, PD-1/PD-L1, tumors, immune escape, regulation

## Abstract

It is obvious that tumor cells have developed a number of strategies to escape immune surveillance including an altered expression of various immune checkpoints, such as the programmed death-1 receptor (PD-1) and its ligands PD-L1 and PD-L2. The interaction between PD-1 and PD-L1 results in an activation of self-tolerance pathways in both immune cells as well as tumor cells. Thus, these molecules represent excellent targets for T cell-based immunotherapies. However, the efficacy of therapies using checkpoint inhibitors is variable and only a limited number of patients receive a long-term response, while others develop resistances. Therefore, a better insight into the constitutive expression levels and their control as well as the predictive and prognostic value of PD-1/PD-L1, which are controversially discussed due to the methodological assessment, the dynamic and time-related variable expression of these molecules, is urgently required. In this review, the current knowledge of the PD-L1 and PD-1 genes, their expression in immune and tumor cells, the underlying molecular mechanisms of their regulation and their association with clinical parameters and therapy responses are summarized.

## 1. Introduction

Recently, immunotherapy has a rekindled interest as an effective therapeutic approach for the treatment of various solid and hematopoietic cancers. However, tumors could efficiently suppress immune responses by activating negative regulatory pathways, also named immune checkpoints (iCP), which are involved in immune homeostasis or adopt features that enable tumor cells actively escaping immune cell elimination [1]. The prototypes and key players of these iCPs are the cytotoxic T lymphocyte protein-4 (CTLA-4) and the programmed cell death protein-1 (PD-1), which are inhibitors of T cell proliferation and function [2,3]. Several studies demonstrated that tumors evade immune surveillance by deregulating survival and proliferation pathways including iCPs. The interaction of PD-1 with its ligands, PD-L1 and PD-L2, provides an immune escape for tumor cells due to inhibition of T cell activation and cytokine production leading to the attenuation of T cell responses by blocking proliferation, apoptosis induction and differentiation of regulatory T cells (Tregs) [4,5]. While the PD-1 receptor is expressed on T cells upon activation during their priming and expansion, *PD-L1* expression is heterogeneous and dynamic in tumors and immune cells. Structural alterations of the *PD-L1* gene as well as its deregulated expression mediated by intracellular and extracellular factors are relevant modulators of *PD-L1* expression.

It has been demonstrated that T cell responses could be stimulated by targeting this pathway with immune checkpoint inhibitors (iCPIs), which block the interaction of PD-1 with its ligands thereby overcoming the intrinsic resistance to immune surveillance by mounting anti-tumor immune responses [6,7,8]. This also leads to an improved outcome and increased survival of patients. However, *PD-L1* expression alone is not sufficient as a predictive factor for stratification of patients responding to immunotherapy [9]. Recently, a link between *PD-L1* expression, tumor mutational burden (TMB) and/or microsatellite instability with the response to iCPIs has been reported in different tumor types, including, e.g., non-small lung cell carcinoma (NSCLC), melanoma, renal cell carcinoma (RCC) and colorectal cancer (CRC) [10,11]. However, despite the fact that PD-1/*PD-L1* blockade therapy has shown remarkable clinical benefit for patients, durable response rates are only found in a minority of patients [12,13]. Therefore, an increased knowledge about the underlying molecular mechanisms that directly or indirectly alter the expression of *PD-L1* is urgently needed to improve the efficacy of PD-1/*PD-L1* directed immunotherapies alone or in combination with other therapies.

## 2. Characteristics of the PD1 Ligands, PD-L1 and PD-L2

PD-L1, also known as B7 homolog1 (B7-H1) with homology to B7-1 and B7-2, PDCD1L1 or cluster of differentiation (CD)274, has been identified as a ligand of the co-inhibitory receptor PD-1 and is encoded by the *CD274* gene localized on chromosome 9p24.1 [14,15]. Under physiologic conditions, it is constitutively expressed in different tissues, but mainly in activated T and B lymphocytes, dendritic cells (DCs), monocytes, mesenchymal stem cells (MSCs), bone marrow (BM)-derived mast cells and various immune privileged organs [16,17,18]. Furthermore, PD-L1 expression could be induced by γ-chain cytokines on T cells [15,19,20] and by IL-21 on CD19^+^ B cells. PD-L1 expression could also be induced by LPS or BCR activation in human B cells and by interferon (IFN)- γ on monocytes as well as on non-lymphoid cells including endothelial cells [19,21,22]. Due to its high evolutionally conserved expression, a functional importance of PD-L1 has been suggested. In addition, PD-L1 is often expressed in the setting of inflammation and/or on tumor cells of distinct origin [23]. In this context, it is also noteworthy, that there exists a broad distribution of PD-L1 in different cellular compartments [24]. These formats include not only membranous PD-L1 (mPD-L1) [25], but also cytoplasmic PD-L1 (cPD-L1) [26,27], nuclear PD-L1 (nPD-L1) [28], serum PD-L1 (sPD-L1) [29] and exosomal PD-L1 [30].

Concerning PD-L1 mRNA expression, two alternative transcripts are generated by the CD274 gene. The long transcript has seven exons with a coding sequence of approximately 800 base pairs (bp) and encodes for a 290 amino acid protein with a molecular weight of 33 kDa. It is a transmembrane glycoprotein and consists of a large extracellular region containing immunoglobulin (Ig)-like domains, a hydrophobic transmembrane domain as well as cytoplasmic tail of 30 amino acids, which does not contain canonical signaling motifs [17]. Exon 1 encodes for the 5′ untranslated region (UTR), whereas exon 7 encodes for a part of the intracellular domain and the 3′ UTR. The second transcript is generated by alternative splicing and absence of the third exon thereby generating a shorter 160 aa isoform of PD-L1 lacking the IgV-like domain.

Furthermore, the PD-L1 promoter has CpG methylation sites with an approximately length of 220 bp suggesting an epigenetic control [31]. It is noteworthy that PD-L1 has a long 3′ untranslated region (3′ UTR) and a number of cis acting elements, which are involved in the mRNA decay. This represents a major determinant of mRNA abundance including an AU-rich element and potential binding sites for microRNAs (miRNA) as well as RNA binding proteins (RBP) [13,32].

The PD-L2 is the second ligand of PD1 sharing approximately 60% amino acid homology in humans to PD-L1 [22]. In contrast to PD-L1, PD-L2 expression is very restricted to activated DCs, macrophages, BM-derived mast cells and peritoneal B1 cells [33]. PD-L2 expression could be induced by LPS and BCR in B cells and by GM-CSF and IL-4 on DCs [22]. Both PD1 ligands are expressed on solid tumors as well as on hematopoietic malignancies, but PD-L2 is expressed to a lesser extent when compared to PD-L1 [34].

## 3. Features of PD-1—Molecular Structure and Expression

The receptor PD-1 (CD279) is a 288 amino acid protein composed of one N-terminal IgV-like domain with a 21%–33% sequence homology to CTLA-4, CD28 and ICOS, a transmembrane domain and a cytoplasmic tail harboring two tyrosine-based signaling domains with an immune receptor tyrosine based inhibitory motif and an immune receptor tyrosine-based switch motif [35]. Constitutive PD-1 expression is expressed on immature CD4^−^ and CD8^−^ thymocytes, activated CD4^+^ and CD8^+^ T cells, but also on B cells, monocytes, NK cells and DCs. It can be also induced on APC, myeloid DC and monocytes via the transcription factors (TF) NFAT, NOTCH, FOXO1 and IRFs, while T-bet represses the transcription of PD-1 [36]. PD-1 expression can be induced by different cytokines, like IL-10 [37], the transforming growth factor (TGF)-β [38] and by chronic infection and cancer. High expression of PD-1 is a characteristic of exhausted T cells [39].

## 4. PD-1/PD-L1-Mediated Signal Transduction and Its Function

The interaction of PD-1 with its ligands has been shown to play an important role in the maintenance of the balance between autoimmunity and peripheral tolerance and impairs anti-viral and anti-tumoral immune responses [13,40]. Since the function of PD-L2 has so far not been elucidated in detail, the review will focus on the PD-1/PD-L1 axis. In contrast to other members of the CD28 family, PD1 transduces only signals when cross-linked together with the B cell or T cell antigen receptor thereby yielding co-inhibitory micro-clusters with the T cell receptor (TCR) and CD28. This results in the inhibition of glucose consumption, cytokine production, proliferation and survival of T lymphocytes. It further prevents the TF expression associated with the effector function including GATA-3, T-bet and Eomes [41]. PD-1 ligation caused a reduced phosphorylation of CD3, ZAP70 and protein kinase C [42], inhibits ERK activation in T cells and the calcium mobilization and phosphorylation of Igβ, Syk, PLC-γ2 and ERK in B cells [43]. Thus, PD-1 ligation attenuates TCR-mediated signaling and impairs the activity of the PI3K-Akt and the Ras/MEK/ERK pathway [44]. One of the mechanisms, by which PD-1 inhibits activation of the PI3K-Akt pathway, includes PTEN phosphorylation and phosphatase activity [45]. This leads to an inhibition of the expression of Skp2 and Cdk2 as well as in increase of p27^kip1^ [46]. The binding of PD-1 to PD-L1 on different cell types contributes to distinct functions. In effector T cells, PD-1 blocks their activity in many ways by inhibiting, e.g., cell cycle progression by affecting multiple regulators of the cell cycle, proliferation, survival and cytokine production [14,15,47]. Furthermore, PD-1 is involved in the TGF-β-mediated signaling of Tregs leading to their proliferation. Ras-PD-1 regulates SWAT3 and synergizes with TGF-β-mediated signals [46]. Next to this, PD-1 ligation alters the metabolic program of activated T cells thereby generating a more oxidative microenvironment [48,49]. PD-1 signaling is well understood and has been recently extensively reviewed [50], while only limited information on PD-L1 signaling is available. However, there exists evidence of a reverse signaling via cross-binding with mAbs.

## 5. Altered PD-L1 Expression in Tumors

PD-L1 is abundantly expressed on various cancer cells [51,52] and has been extensively summarized in various reviews [13,53]. The discrepancy of some results might be due to inadequate sampling of the tumor and insufficient sensitivity of the detection method/antibody used for immunohistochemistry (IHC). Therefore, strong efforts are currently performed to harmonize the staining of PD-L1 [54]. Next to the analytical variability, PD-L1-specific IHC might not reflect the tumor status due to individuality and tumor heterogeneity [55]. The intra-tumoral heterogeneity is common, varying temporal and in scale and was also present between primary tumors and metastases [56]. PD-L1 is also expressed on non-tumor cells of the tumor microenvironment (TME) including macrophages, myeloid DC, MDSC and fibroblasts [57]. Based on its expression pattern and its association with clinical parameters like, e.g., prognosis and patients’ survival [58], PD-L1 expression has been suggested as a biomarker for PD1/PD-L1 inhibitor therapy and response. It must also be taken into account that tumor cells can secrete a vast majority of PD-L1 via exosomes, which promote tumor growth by suppressing T cell function in vitro and in vivo [59]. It is critical to understand how the PD-L1 expression levels are generally regulated and whether this regulation is distinct in immune and tumor cells. Therefore, the highly complex regulation of PD-L1 summarized in Table 1 will be discussed in the following sections.

## 6. Genetic Alterations of PD-L1

### 6.1. Structural Alterations of PD-L1—Loss and Gain

So far, the underlying molecular mechanisms that govern the expression levels of *PD-L1* are not well understood, but could be caused by genetic abnormalities. Over the last years, aberrant *PD-L1* expression in tumors and immune cells could be due to structural alterations, such as DNA copy number alterations (CNA) mediated by gene deletions or gene amplifications as well as translocations. In large B cell lymphoma, a translocation was found in 20%, an amplification in 29% of cases [60]. Interestingly, deletions of *PD-L1* were more frequently found than gains and were most prominent in melanoma as well as NSCLC [61,62]. In many cancer types, copy number gains (CNG) were found in focal regions, chromosome 9p24.1 or in the whole of chromosome 9, which correlated with increased *PD-L1* expression levels and response to iCPI therapy as determined by analysis of datasets from The Cancer Genome Atlas (TCGA) of different cancer types [63]. The frequency of CNG varied between 7% and 14% depending on the tumor entity analyzed and were detected in head and neck squamous cell carcinoma (HNSCC), CRC, cervical, bladder and lung cancers as well as in sarcomas [64,65,66,67,68]. PD-L1 amplification and deletion were associated with TMB evaluated by next generation sequencing based comprehensive profiling [69,70]. However, PD-L1 amplifications did not always correlate with high positive PD-L1 expression determined by IHC [71]. PD-L1 CNGs were in particular found in NSCLC of smokers with mutations and EML4-ALK rearrangements [72]. In NSCLC, CD274 amplifications associated with an increased PD-L1 expression, which co-occurred with Janus Kinase (JAK)2 amplifications [73]. In cervical, lung and squamous cell cancers, PD-L1 amplification was frequently accompanied by an amplification of the *PD-L2* gene [68], while in triple negative breast cancer (TNBC), genomic amplification of 9p24.1 was found [74,75]. It is noteworthy that PD-L1 amplification was present primary tumors and associated metastasis with a concordance rate of 73%.

### 6.2. PD-1 and CD274 Polymorphisms

There exist only little data on the impact of PD-L1 polymorphisms on PD-L1 expression. The single nucleotide polymorphism (SNP) rs4143815C>G in the 3′ UTR of PD-L1 has been associated with an increased cancer risk and might be used as biomarker to predict the susceptibility to cancer [76]. Two polymorphisms characterized by >G changes localized in the 5′ UTR of PD-L1 serve as binding site for SP1 [77] thereby affecting PD-L1 expression. Thus, variations in the PD-L1 promoter could lead to an overexpression. In addition, a number of polymorphisms were also described in PD-1, which were associated with the risk of susceptibility of different cancers [78].

## 7. Control of PD-L1 Expression

### 7.1. Control of PD-L1 Expression by Aberrant Oncogenic Signaling

The upregulation of PD-L1 expression could be a consequence of constitutive oncogenic signaling in tumor cells. Indeed, overexpression of the MYC oncogene detected in approximately 70% of cancers [79] leads to an upregulation of PD-L1 expression mediated by binding of MYC to the PD-L1 promoter [80]. Vice versa inhibition of MYC results in downregulation of PD-L1 expression [80,81]. Signaling via the PIK3/AKT/mTOR pathways also control immune surveillance, while PTEN loss and mutations in PIK3CA induce the activation of the AKT/mTOR pathway thereby increasing PD-L1 expression [82], which could be associated with immune resistance [45]. The EGF-R expression is a strong independent predictive marker of PD-L1 overexpression, while inhibition of EGF-R expression using tyrosine kinase inhibitors (TKI) leads to a downregulation of PD-L1 expression [83,84]. Thus, PD-L1 expression can be induced by EGF signaling and enhanced by activating mutations in the EGF-R gene and contributes to the EGF-R-driven immune escape [85]. In addition, the Ras/Raf/MEK/MAPK-ERK pathway was also linked to PD-L1 overexpression in various cancers, while respective inhibitors lead to a downregulation of PD-L1 expression [86,87]. PD-L1 expression could be upregulated by cell contact. Furthermore, a cell-dependent juxtacrine signaling via the membrane receptor tyrosine kinase ephrin receptor A10 (EphA10) was demonstrated to upregulate PD-L1 expression [88].

### 7.2. Transcriptional Regulation of PD-L1 Expression by Inflammatory Signals

PD-L1 expression is induced by local inflammatory signals as well as by interferon (IFN)-γ produced by T lymphocytes in multiple tumor types [51], but also on healthy tissues and immune cells, but the effect of IFN-γ on PD-L1 expression is context-dependent. The inhibition of the NF-κB pathway impairs the IFN-γ-mediated induction of PD-L1 expression [89], which is in line with the inhibition/downregulation of PD-L1 expression by inhibitors directed against NF-ĸB, JAK1/JAK2 and BTK [90]. In addition to IFN-γ, type I IFNs can also induce PD-L1 expression [91]. Other inflammatory stimuli affecting PD-L1 expression in tumors include lipopolysaccharide (LPS) by triggering TLR4, TNF-α and IL-4 and in DC, in particular IL-1β, IL-6, IL-27, IL-10 and TGF-β [13] (summarized by Sun et al., 2018). Furthermore, chemotherapeutics, such as anthracycline and taxane up-regulate PD-L1 expression in some subtypes of triple negative breast cancer (TNBC) via NF-_K_B [92]. Next to the transcriptional control of PD-L1 expression, epigenetic mechanisms could lead to altered expression.

### 7.3. Epigenetic Deregulation of PD-L1 in Tumors

Epigenetic alterations such as promoter DNA methylation and histone modifications play an important role in the regulation of PD-L1 expression [31,93]. As already mentioned, a CpG island has been identified in the 5′ UTR of the PD-L1 in gene. Hypermethylation of the CD274 promoter leads to a reduced/lack of PD-L1 expression, which could be reverted by the treatment of cells with the demethylating agents 5-azacytidin (5-AZA), which increased the expression of PD-L1 expression at the mRNA and protein level [94]. In acute myeloid leukemia (AML), this was accompanied by host dependent upregulation of not only PD-L1, but also PD-1 mRNA [95]. Molecular analyses demonstrated the high density of methylation at the CpG islands in the promoter region. Hypermethylation of the PD-L1 promoter was correlated with lower PD-L1 expression, while a strong hypomethylation was significantly associated with increased expression of CTLA-4 and PD-1. Thus, PD-L1 promoter methylation may provide a potentially more effective immunotherapeutic strategy in some tumor patients. Since PD-L1 overexpression transiently occurs during the cytokine-driven epithelial mesenchymal transition (EMT), a strong link between PD-L1 promoter demethylation and the TGF-β signaling pathway was found, which was associated with the loss of DNA methyltransferase 1 (DNMT1) in lung cancer cells [96].

### 7.4. Disruption of the PD-L1 3′ Untranslated Region

The 3′ UTR of many genes is important in the mRNA decay resulting in altered protein expression levels. Structural alterations leading to the disruption of the 3′ UTR of the PD-L1 gene have been recently described [97] and appear to be highly common in different cancer types including T cell leukemia/lymphoma, diffuse large B cell lymphoma and stomach adenocarcinoma. Furthermore, disruption of the PD-L1 3′ UTR in mice enables immune evasion of tumor cells with elevated PD-L1 expression, which is actively inhibited by immune checkpoint blockade. Abnormalities in the 3′ UTR could on the one hand induce a stabilization of PD-L1 transcripts, while the 3′ UTR has also potential microRNA (miRNA) binding sites, which are involved in intra-tumoral immune suppression. Indeed, a mutation in the 3′ UTR has been shown to disrupt the miR-570 binding thereby enhancing PD-L1 expression [98]. The lack of the C-terminal region of PD-L1 could induce variations in the efficacy of different anti-PD-L1 antibodies in protein detection.

### 7.5. MicroRNAs Targeting PD-L1

MicroRNAs (miRNAs) are small non-coding RNAs of about 20 nucleotides in length and represent next to RNA binding proteins (RBPs) the main posttranscriptional regulators of gene expression by binding to the 3′ UTR of target genes, but also to coding sequences [99]. MiRNA expression is deregulated in human cancers and function as oncogene and tumor suppressor genes. Recently, an increasing number of studies have identified miRNAs with an immunomodulatory potential [100]. For the identification of PD-L1 specific miRNAs, different approaches were used, such as mRNA prediction tools, RNA sequencing as well as miTRAP in combination with RNAseq. A number of miRNAs targeting the 3′ UTR of PD-L1 were found in different murine and human tumor models leading to a downregulation of PD-L1 expression, which has been recently summarized [13]. This was in some cases associated with an inhibition of tumor growth, invasion, cell migration and EMT as well as enhanced T cell responses and reduced chemoresistance. In contrast to the inhibition mediated by directly binding of miRNAs to the 3′ UTR of PD-L1 thereby contributing to an altered transcriptional and posttranscriptional regulation, there exist little information about the indirect modulation of miR-mediated signaling networks driving PD-L1 expression. MiR-197 is a negative modulator of PD-L1 expression, which is mediated via the CKS1B-STAT3 cascade in association with the overexpression of various genes like cyclin D1, survivin, c-myc and Bcl2, in lung and oral squamous cancer [101,102]. MiR-20, miR-21 and miR-130b repress the expression of PTEN, which results in an increased PD-L1 expression. So far, RBPs binding to the 3′ UTR of PD-L1 have not yet been identified, but have been suggested to be also involved in the regulation of PD-L1 expression. In contrast, deregulation of PD-L1 by long-noncoding RNAs (lnRNAs) has been reported [103]. For example, the lncRNA small nucleolar RNA host gene SNHG20 modulates the expression of PD-L1 and JAK2 in esophageal squamous cell carcinoma. Others include the lncRNAs HOTTIP [104], SNHG14 [105] and MALAT1 [106,107].

### 7.6. Posttranslational Modifications of PD-L1

Posttranscriptional regulation of PD-L1 expression has been recently explored by various researchers and could be mediated by protein glycosylation, phosphorylation, ubiquitination, sumoylation acetylation as well as lysosomal degradation [108,109,110]. Alterations of these post-translational modifications (PTMs) might influence the PD-L1 mediated immune resistance and might represent potential targets for enhancing anti-tumor immune responses.

PD-L1 is highly glycosylated resulting in heterogeneous protein expression patterns [111]. The CD33 kDa form represents the non-glycosylated PD-L1 protein, while upon glycosylation the MW of PD-L1 ranges between 45 and 55 kDa. Inhibitors of N-linked, but not O-linked glycosylation alters the migratory shift of PD-L1 suggesting that PD-L1 is primarily N-glycosylated. The glycosylation leads to a stabilization of PD-L1 expression in tumor cells including cancer stem cells with a 4-fold longer protein half-life [111], which is associated with escape from immune surveillance. Thus, a direct targeting of PD-L1 glycosylation might be an effective strategy to improve iCPI therapy [110]. Indeed, TNBCs were eliminated by targeting glycosylation [112]. Furthermore, the non-glycosylated PD-L1 can be degraded by the ubiquitin proteasome system after phosphorylation. Thus, targeting of the PD-L1 poly-ubiquitination might be an alternative approach to enhance immune checkpoint therapy. PD-L1 is subjected to acetylation tyrosine-phosphorylation as well as mono-ubiquitination upon EGF stimulation. Thus, PTMs have emerged as an important aspect in the PD-L1 regulation. However, so far it is not clear, how PTMs influence the subcellular localization of PD-L1 thereby contributing to the function of intracellular PD-L1 expression.

Several regulators of PD-L1 expression were described. CMTMG, a type-3 transmembrane protein, was identified to control PD-L1 expression in different tumor cells and in LPS-stimulated DC [108,113]. It associates with the PD-L1 protein leading to a reduced ubiquitination and on increased PD-L1 protein half-life. Furthermore, CDK4 is a negative regulator of PD-L1 function by indirectly regulating its ubiquitination [114]. Another possible interaction between PD-L1 and the glycogen synthetase kinase 3b was described leading to an increased PD-L1 degradation [111].

### 7.7. State of the Art PD-1/PD-L1 Based Immunotherapy and Therapy Resistances

Blockade of the PD-1/PD-L1 axis with the anti-PD-1 antibodies pembrolizumab and nivolumab as well as the anti-PD-L1 antibodies atelizumab, avelumab and durvalumab has been approved for the treatment of various cancers. This includes melanoma, NSCLC, RCC, HNSCC, urothelial and hepatocellular carcinoma, TNBC, CRC and gastric cancer. Merkel cell carcinoma and Hodgkin’s lymphoma as standard therapy and/or alone and in combination with other therapies in clinical phase II and III trials, which have been recently summarized in a number of reviews [115,116,117]. Despite the complete, partial and overall response rate highly varied between the tumor types analysed and iCPI employed, the progression-free and the overall survival was improved in almost all subgroups of patients independent of PDL1 expression. However, there are patients that are not benefitting from this treatment, including patients with tumors carrying EGF-R mutations [118]. Similar holds for the use of PD-L1 expression as a predictive marker to guide the iCPI treatment. Thus, the identification of novel predictive biomarkers for iCPI therapy is urgently required, since the heterogeneity of PD-L1 expression in tumors could also have significant implications for its accuracy as predictive marker and for therapy response [56,119]. This may include HLA class I antigens and components of the antigen processing as well as type I and type II IFN pathways [120]. Defects in these molecules and an increased canonical signaling have recently been shown to be associated with intrinsic and/or acquired resistance to PD-1/PD-L1 inhibition [121,122,123,124]. A systematic meta-analysis revealed that a high TMB could also predict the improved efficacy of iCPI in cancers suggesting that targeted next generation sequencing for estimating TMB might be standardized to eliminate heterogeneity [125,126].

## 8. Conclusions

Despite PD-L1 expression has been determined in a vast number of tumors of distinct origin and correlated to clinical parameters and response to PD-1/PD-L1 iCPI, its value as biomarker for prognosis and therapy response is still discussed. Therefore, a better understanding of the underlying mechanisms of its heterogeneous expression is required. In-depth analysis of molecular processes revealed a complex control of PD-L1 expression, which varies from structural abnormalities to deregulation mediated by transcriptional, epigenetic, post-transcriptional and posttranslational level. This increased knowledge might also offer novel opportunities for the treatment of tumors by targeting, e.g., respective signal pathways, PTMs or epigenetic modifications.

## Figures and Tables

**Table 1 jcm-08-02168-t001:** Distinct mechanisms regulating PD-L1 expression.

copy number alterations/amplifications/translocation
transcription factors
oncogenic pathway activation
cytotoxic agents/chemotherapeutics
miRNAs and lncRNAs
posttranslational modifications

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
