# Peer review of "Basis of PD1/PD-L1 Therapies"

_jcm, 2019, doi:10.3390/jcm8122168_

Round 1

Reviewer 1 Report

An excellent, comprehensive and timely  review of the molecular mechanisms surrounding  PD1/PDL1 regulation of expression.

As a relatively minor deficiency the manuscript would have been strengthened by a brief description of other factors that act a prognostic and predictive factors related to antagonism of the  PDL-1 axis ie, HLA, TME and expanded TMB discussion. In addition  tumour heterogeneity of PDL-1 tumour expression within a tumour nodule (ie, clonal diversity ) itself should be expanded upon  as it has possible implications to clinical response and choice of therapy.

In addition to the nice discussion of constitutive pathway action and resultant PDL-1 expression ie, EGFR activating mutations in NSCLC it would have  been found helpful to include a discussion around this expression and it's  impact  or lack thereof on PDL-1 axis signalling as it pertains to clinical antagonism.

Author Response

Reviewer 1: Point 1: we included some predictive factors (EGFR mut,
HLA, IFN signaling) and added also some information regarding the TMB. Point 2: We included NSCLC lit and data regarding PD-L1
and EGFR mutations.

Reviewer 2 Report

This is in principle an interesting review that will benefit of some implementation. First and most important, I believe that a short para summarizing the state of the art in PD-1/PD-L1-base therapy would be valuable. In other words, i suggest that the Author provides additional info on the rate of success/insuccess of PD-1/PD-L1-targeted therapies. These data are easily accessible in the literature.

Secondly, I do not understand the Table reported at pg 7 (Table 1). What does it mean?

Thirdly, I do not understand why two Reference sections are shown: Literature (line 320) and References (line 480).

In the fourth place, usage of English language should be thoroughly revised (e.g. lines 114-115; 136-137; 148 and others). Incidentally, what is proteinase C (line 130)?

Finally, the average lay reader loves to see schematic renditions (figures) that better convey the key message of the review.

Author Response

Reviewer 2: Point 1: We included some information on the success rate and
failure of iCPI treatment in tumor patients, but since there
exist a number of recent reviews we just cited mainly the
literature. In addition, it is noteworthy that the aim of
this report was to described and focus on the biology of
PD-1/PD-L1, the PD-L1/PD-1 immunotherapy is subject of
another review in this issue from another author. Point 2: Table 1 shows the mechanisms involved in modulating
PD-l1 expression in tumor and immune cells and summarizes
the information given in the manuscript. Point 3: The first reference section was deleted, sorry for
the mishap. Point 4: We revised the English in the parts as requested,
these mistakes were overlooked in the last version.

Round 2

Reviewer 2 Report

None